# Biology of Two-Spotted Spider Mite (*Tetranychus urticae*): Ultrastructure, Photosynthesis, Guanine Transcriptomics, Carotenoids and Chlorophylls Metabolism, and Decoyinine as a Potential Acaricide

**DOI:** 10.3390/ijms24021715

**Published:** 2023-01-15

**Authors:** Ambra S. Parmagnani, Giuseppe Mannino, Carla Brillada, Mara Novero, Luca Dall’Osto, Massimo E. Maffei

**Affiliations:** 1Department of Life Sciences and Systems Biology, University of Turin, Via Quarello 15/A, 10135 Turin, Italy; 2Albert-Ludwigs Universität Freiburg, Institut für Biologie II, Zellbiologie, Schänzlestr. 1, D-79104 Freiburg, Germany; 3Laboratory of Photosynthesis, Department of Biotechnology, University of Verona, Strada Le Grazie 15, 37134 Verona, Italy

**Keywords:** *Phaseolus lunatus*, spider mite, guanine, RubisCO, carotenoids, chlorophylls, acaricide, photosystems I and II, confocal laser scanning microscopy, chloroplasts

## Abstract

Two-Spotted Spider Mites (TSSMs, *Tetranychus urticae* Koch 1836 (Acari: Tetranychidae)) is one of the most important pests in many crop plants, and their feeding activity is based on sucking leaf cell contents. The purpose of this study was to evaluate the interaction between TSSMs and their host Lima bean (*Phaseolus lunatus*) by analyzing the metabolomics of leaf pigments and the transcriptomics of TSSM guanine production. We also used epifluorescence, confocal laser scanning, and transmission electron microscopies to study the morphology and structure of TSSMs and their excreta. Finally, we evaluated the potential photosynthetic ability of TSSMs and the activity and content of Ribulose-1,5-bisphosphate Carboxylase/Oxigenase (RubisCO). We found that TSSMs express several genes involved in guanine production, including Guanosine Monophosphate Synthetase (*GMPS*) and decoyinine (DCY), a potential inhibitor of GMPS, was found to reduce TSSMs proliferation in infested Lima bean leaves. Despite the presence of intact chloroplasts and chlorophyll in TSSMs, we demonstrate that TSSMs do not retain any photosynthetic activity. Our results show for the first time the transcriptomics of guanine production in TSSMs and provide new insight into the catabolic activity of TSSMs on leaf chlorophyll and carotenoids. Finally, we preliminary demonstrate that DCY has an acaricidal potential against TSSMs.

## 1. Introduction

The two-spotted spider mite (TSSM, *Tetranychus urticae* Koch 1836 (Acari: Tetranychidae)), a polyphagous species that feeds on more than 1100 host plants, sucks the plant cell content of leaf mesophyll. TSSM is one of the most important pests in many cropping systems worldwide and the most polyphagous species within the family of Tetranychidae [1]. TSSM feeding leads to severe leaf damage, reduction in photosynthetic capacity, and finally, leaf abscission [2]. TSSM interacts with the host plants by secreting elicitors (the so-called tetranins) that trigger early and late plant responses, including the activation of genes involved in direct and indirect defense [3]. The digestive system of spider mites consists of the foregut (mouth, pharynx, esophagus), midgut (ventriculus and coeca), and hindgut (excretory organ, rectum, and anus) [2,4]. Processed food is eventually compacted in the excretory organ and excreted as fecal pellets of frass [5]. In addition to excreting chlorophyll-degradation by-products, the excretory organ is known to function in the excretion and removal of nitrogenous metabolic wastes by producing guanine [6]. Chlorophyll and chlorophyll degradation products have been observed in the two-spotted spider mite revealing distinct patterns of chlorophyll degradation and guanine formation [5,7]. The process of guanine metabolism still remains enigmatic in many respects because there are direct evidence of its formation, transport, and secretion by the epithelial cells of the excretory organ [8,9]. In arachnids, biogenic guanine crystals are not pure crystals but molecular alloys made of solid solutions and mixed crystals of guanine, hypoxanthine, and sometimes xanthine [8]. Guanine crystals are the most widespread organic biocrystal, responsible for an enormous array of optical phenomena associated with animal coloration and vision. Biogenic guanine crystals are constructed from π-stacked, H-bonded molecular layers [10]. At least nine distinct guanine morphologies are known; guanine crystals form intracellularly in membrane-bound vesicles (the “crystal vesicle”) in specialized cells called iridophores or guanocytes. In spiders, guanine crystallization is a “non-classical,” multi-step process involving a progressive ordering of states [10].

Although the catabolic fate of food products and the development of spider mites have been investigated on numerous occasions [9,11,12], to our knowledge, there are a few reports on the metabolisms and gene expression of guanine and metabolism of chlorophyll and carotenoids in spider mites.

Another important aspect of TSSM’s biology is their ability to rapidly develop resistance to chemicals [13]. The frequent application of acaricides, combined with the short life cycle and high reproductive potential of TSSMs, results in the rapid development of resistance [1,14]. TSSM was shown to be resistant to several acaricides, including organophosphates, pyrethroids, bifenazate, Mitochondrial Electron Transport Inhibitors (METIs), ketoenols, chlorfenapyr, dicofol and mite growth inhibitors (such as clofentezine, hexythiazox and etoxazole) ([1] and references therein). Therefore, it is important to understand the biochemical and molecular mechanisms involved in spider mite metabolism. In this context, decoyinine (DCY) or 9-(6-deoxy-β-D-erythro-hex-5-enofuran-2-ulosyl) adenine, also known as angustmycin A, is an analog of adenosine which causes a decrease in intracellular GTP levels by inhibition of GMP synthetase, by impairing the conversion of xanthosine monophosphate (XMP) to GMP and by interfering with pyrophosphokinase activity [15,16]. Research on DCY has recently focused on its applications in botany and agronomy [17]. For instance, rice seed priming with DCY increased the survival rates of rice seedlings under heavy infestation by the brown planthopper (*Nilparvata lugens*, BPH) by making it difficult to use the intercellular stylet pathway to reach the phloem [18], whereas seed priming with DCY can confer rice resistance to BPH [18] and lower the fecundity of small brown planthopper (*Laodelphax striatellus*, SBPH) by 40.27%. DCY significantly altered the rice activities of antioxidant enzymes (CAT, SOD, and POD) and synthases for secondary metabolites (PAL and PPO), MDA, H_2_O_2,_ and soluble sugars against SBPH infestation [19].

The aim of this work was to describe the genomics of TSSMs guanine formation and the metabolomics of chlorophyll and carotenoids. We also tested the ability of DCY as a potential acaricide against TSSMs infestation. 

## 2. Results

### 2.1. Fluorescence Microscopy Differentiates Chlorophyll and Guanine Metabolism in the TSSM

The localization of chlorophyll and its degradation products and guanine formation can be visualized by microscopy inside the TSSMs and in their excreta. When observed under a bright field, TSSMs show the characteristic two red pigment spots and the typical spots in the two lateral caeca (Figure 1A). By turning on simultaneously bright light and UV light, the leaf mesophyll chlorophyll and the chlorophyll inside the lateral caeca show their characteristic red fluorescence, whereas the TSSM cuticle and the excretory organ show a bright bluish fluorescence (Figure 1B). By using the fluorescence setting for DAPI localization, we observed in TSSMs the lateral caeca, excretory organ, and their excreta, a clearer distinction between the red chlorophyll fluorescence organized in individual agglomerates (Figure 1C, double arrow) and the presence of guaninesomes with green fluorescence (Figure 1C, single arrow). By using confocal laser scanning microscopy (CLSM), we confirmed the distinction between chlorophyll degradation products (Figure 1D, double arrow) and the typical fluorescence of guaninesomes. Images of TSSMs excreta (pellets) obtained by epifluorescence microscopy show chlorophyll degradation products with different levels of red fluorescence (Figure 1E, double arrow) and guaninesomes (Figure 1E, single black arrow) characterized by a bright fluorescence. When observed under CLSM, pellets are characterized by a red fluorescence which is associated with chlorophyll degradation products (Figure 1F, double arrow), while guaninesomes are differentiated by a green fluorescence (Figure 1F, single arrow).

### 2.2. TSSM Ultrastructure Reveals the Presence of Intact Chloroplasts

It is known that the polyphagous TSSM feeds on plants by sucking the plant cell content of leaf mesophyll. Ultrathin sections of TSSMs observed under the transmission electron microscope revealed the presence of intact chloroplasts located in the mite lateral caeca. In particular, intact chloroplasts could be clearly distinguished just below the TSSMs cuticle (Figure 2A,B, Cu). Chloroplasts showed an intact thylakoidal systems with both stromal and stacked thylakoids (Figure 2A, T) still organized in grana (Figure 2B, G). Chloroplasts were found to be in close proximity to mitochondria and nuclei (Figure 2A, M, and N). Inside the lateral caeca, chloroplasts were found with intact thylakoids with both stromal and grana lamellae (Figure 2C, G), along with the presence of plastoglobules (Figure 2C, P). Inside the lateral caeca, the sectioning of a pellet corresponding to the red fluorescent pellets indicated with a double arrow in Figure 1D shows a vacuolized structure with remnants of the chloroplast degradation (Figure 2D).

### 2.3. Guanine Is Excreted by TSSM

The chemical nature of the green fluorescent particles was assessed by HPLC-MS analysis of both TSSMs and their excreta. We confirmed the presence of guanine, as shown by the mass spectrum reported in Figure 3.

### 2.4. Guanine Is Produced by TSSMs through the Upregulation of Most of the Guanine Biosynthesis Coding Genes

The guanine biosynthetic pathway is illustrated in Figure 4. The first two steps involve the transformation of α-d-ribose-5-pyrophosphate (R5P) to 5-phosphoribosyl-β-amine (5-PRA) and are catalyzed by enzymes coded by 5-phosphoribosyl-1-pyrophosphate synthetase (*PRPS1*) and glutamine phosphoribosyl pyrophosphate aminotransferase (*PPAT*). The addition of glycine and *N*^10^-formyl tetrahydrofolate (THF) in the presence of ATP yields formylglycinamide ribonucleotide (FGAR) by catalysis of an enzyme coded by glycinamide ribonucleotide formyltransferase (*GART*). FGAR is then transformed to formylglycinamidine ribonucleotide (FGAM) by the transcript of phosphoribosyl formylglycinamidine synthase (*FGAMS*). A series of reactions leads to *N*-succinyl-5-aminoimidazole-4-carboxamide ribonucleotide (SAICAR), in which the aspartate addition in the presence of ATP is catalyzed by phosphoribosylaminoimidazolesuccinocarboxamide synthase, coded by the *PAICS* gene. Adenylosuccinate lyase (coded by the gene *ADSL*) removes fumarate yielding 5-aminoimidazole-4-carboxamide ribonucleotide (AICAR), which is then transformed to inosine monophosphate (IMP) by a series of reactions the last of which is catalyzed by AICAR transformylase/IMP cyclohydrolase coded by the *ATIC* gene. Oxidation of IMP to xanthosine monophosphate (XMP) is catalyzed by IMP dehydrogensase (coded by *IMPDH1*), whereas the transamination of XMP to guanosine monophosphate (GMP) is catalysed by GMP synthetase, which is coded by the *GMPS* gene. Finally, the sequential removal of phosphate and ribose (the latter catalyzed by Purine nucleotide phosphorylase coded by the *PNP* gene) yields guanine.

We then assessed the expression of the genes encoding for the enzymes involved in these key steps of guanine biosynthesis in the TSSMs (Figure 5). Most of the genes were significantly (*p* < 0.05) upregulated, with the sole exception of the genes involved in the transformation of 5-PRA to FGAM, where *GART* was significantly (*p* < 0.05) downregulated and *FGAMS* showed no significant difference (*p* = 0.48) with respect to the housekeeping gene expression. In particular, a strong upregulation was found for *GMPS*, which codes the crucial step of the transformation of XMP to GMP (Figure 5).

### 2.5. Decoyinine Inhibits GMPS and Reduces the TSSM Damage to Lima Bean Leaves

Decoyinine (DCY, Figure 6) causes a decrease in intracellular GTP levels by inhibition of *GMPS* and impairs the conversion of XMP to GMP by interfering with pyrophosphokinase activity [15,16]. Because we observed that one of the most upregulated genes in TSSM guanine synthesis is *GMPS*, we evaluated the possibility of using DCY as a potential acaricide.

Preliminary results indicate that the application of DCY significantly (*p* < 0.05) reduced the leaf damage inferred by TSSMs to Lima bean leaves and that this reduction was as a function of DCY concentration, and that a higher activity was observed at the beginning (4th day) of infection (Figure 7).

### 2.6. TSSMs Contain Mainly Pheophytin, and Their Feeding Activity Decreases Chlorophyll and Increases Pheophytin Contents in Lima Bean Leaves

To investigate whether the interaction with TSSMs could produce changes in the pattern of photosynthetic pigments of Lima bean leaves, the chlorophyll (Chl) composition was evaluated (Figure 8). In healthy Lima bean leaves, analyses revealed that Chl a and Chl b were the main photosynthetic pigments, along with their respective epimers (Chl a′ and Chl b′) (Figure 8A). Although the epimers have the same spectral profile as their parents, they can be further discriminated. Indeed, Chl a′ and Chl b′ elute after the corresponding Chl a and Chl b forms [20]. Both pheophytin (Pheo) a and b and their epimers were also detected (Figure 8, Appendix A).

In general, the highest total amount of Chl and Pheo was found in healthy leaves, followed by leaves damaged by TSSMs and by the content analyzed in the TSSMs (Table 1). Healthy leaves contained more than 65% Chl a and about 17% of Chl b, displaying a 3.8 Chl a/b ratio. The Chl corresponding epimers (Chl a′ and Chl b′) were also detected, but in lower amounts (Table 1). Chl a′ accounted for 17%, and Chl b′ for 1% only. Other Chl degradation products were present in very low amounts accounting for less than 1% of the total Chl content (Table 1). Pheo b and Pheo b’ were not detected, whereas the content of Pheo a and Pheo a’ was below 0.6% of the total Chl and Pheo content (Table 1 and Figure 8A).

In Lima bean leaves infested by TSSM, both Chl a and Chl b and their respective epimers decreased with a concomitant rise in the Pheo contents (Pheo a′, Pheo a, Pheo b′, and Pheo b) with respect to healthy leaves (Table 1 and Figure 8B). In particular, Chl a was lowered to 54%, while Chl b was 13% of the total Chl content, thus explaining a significantly (*p* < 0.05) higher Chl a/b ratio (4.1) with respect to healthy leaves. The respective epimers also decreased to 8% and 0.55% for Chl a′ and Chl b′, respectively. However, with respect to healthy leaves, there was an increase in both Pheo a and Pheo b and their respective epimers Pheo a′ and Pheo b′ (Table 1 and Figure 8B).

Qualitative analyses performed on TSSMs, revealed a significant reduction in photosynthetic pigments within the organism, with a drastic increase in Pheo content (Table 1 and Figure 8C). Specifically, Pheo a and Pheo b contributed 76% and 13% of the total Chl and Pheo content, respectively, while Chl a and Chl b contributed less than 10% (Table 1). There was also a significant decrease in the Chl a/b ratio (1.7) with respect to both healthy and TSSM-infested leaves (Table 1).

### 2.7. TSSMs Contain a High Content of Carotenoids, and Their Feeding Activity Decreases the Total Carotenoid Content of Lima Bean Leaves

Regarding the carotenoid composition, nine different compounds were detected. Figure 9 shows the chromatogram derived from HPLC analysis of the metabolites extracted from healthy leaves (Panel A), TSSM-infested leaves (Panel B), and TSSMs (Panel C). In particular, with regards to carotenes, both *cis*- and *trans*-α-carotene were identified, along with 15-*cis*-β-carotene, 13-*cis*-β-carotene, 9-*cis*-β-carotene, γ-carotene, and *trans*-β-carotene. Moreover, also the xanthophylls lutein and an unidentified compound, probably belonging to the same xanthophyll class, were observed (Figure 9, Appendix A). Finally, the chromatogram of the TSSM sample (Figure 9C) shows two peaks (marked with #) originating from contamination of Pheo a’ and Pheo a.

Although HPLC-DAD did not reveal qualitative changes in the carotenoid profile between healthy and TSSM-infested leaves, quantitative differences were observed (Table 2). Particularly, TSSM-infested leaves generally showed a significantly (*p* < 0.05) lower carotenoid content than healthy leaves. The compound with the highest decrease was lutein, whose content was decreased by more than 70%. In contrast, minor decrements, ranging between 20% and 30%, were found for all other carotenoids and xanthophylls. The only compound that remained statistically unchanged was a putative xanthophyll. Regarding TSSM, the total carotenoid content was four times higher than the one measured in healthy leaves and TSSM-infested leaves. Interestingly, despite the higher values recorded for TSSMs, two compounds, lutein, and 9-*cis*-β-carotene, were not detected in TSSMs (Figure 9, Table 2).

### 2.8. TSSMs Feeding Activity Alters the Protein and Sugar Content of Lima Bean Leaves

The total content of proteins in healthy Lima bean leaves was 1.93 (±0.04) mg g^−1^ f.wt. and was significantly (*p* < 0.05) increased to 2.10 (±0.04) mg g^−1^ f.wt. after TSSMs feeding. The protein content was higher in the TSSMs 3.02 (±0.09) mg g^−1^ f.wt. We also analyzed the total sugar content of leaves before and after TSSMs feeding. Healthy leaves had a total sugar concentration of 18.26 (±1.80) mg g^−1^ f.wt, which was significantly (*p* < 0.01) lower than in TSSM-infested leaves (24.54 ± 2.11 mg g^−1^ f.wt).

### 2.9. Do TSSMs Retain Photosynthetic Activity?

The presence of intact chloroplasts inside the lateral caeca of TSSMs (Figure 2) and the occurrence of chlorophyll inside the TSSMs (Table 1) raised the question of whether these chloroplasts may retain photosynthetic activity to support the TSSM metabolism.

Upon excitation of Chl a at 440 nm, the fluorescence emission spectra of chloroplasts from Lima bean leaves showed three peaks at 680, 685–692, and 735 nm (Figure 10A, left panel) from LHCII antenna proteins, PSII cores, and PSI-LHCI complexes, respectively. As shown in Figure 10A, right panel, the spectra of TSSM extracts lacked the emission peaks characteristics of PSII core while revealing a small emission peak at 680 nm, indicating that at least part of the LHCII antenna existed as free complexes in the TSSM. Moreover, major emission components peaking at ~713–724 nm were detected, which might originate from either the PSI core devoid of its antenna system or denatured/aggregated PSII subunits. These results suggest that most of the Chl-proteins detected in TSSM extracts are not part of functioning photosystems.

Further insights into the electron transport activity of the PSII were obtained by analyzing the typical 0JIP chlorophyll fluorescence transients, measured in vivo after starting actinic light. In dark-adapted leaves, all reaction centers are in an “open” state, and the electron transport chain between photosystems is completely oxidized; this corresponds to the minimal fluorescence quantum yield of the system. Upon switching on actinic illumination, charge separation occurs in photosystems, and in PSII, the electrons move to Q_A_ > Q_B_ > plastoquinone acceptors. Gradual closure of centers increases fluorescence quantum yield, and when the majority of Q_A_ molecules is reduced, the “J” state of the fluorescence kinetics has been reached. The complete reduction in Q_B_ and of the plastoquinone pool correspond to the “I” and the “P” level of fluorescence quantum yield, respectively (Figure 10B). TSSMs showed no fluorescence transient, thus confirming the lack of functional PSII centers in the sample.

In order to monitor in vivo the photosynthetic electron flow through PSI during actinic treatment, we estimated the redox state of reaction center P700 by measuring absorbance changes at 705 nm. P700 of leaves was rapidly oxidized to P700^+^ under far-red illumination, which favors PSI photochemistry, then reduced in the dark (Figure 10C). The far-red light-induced absorbance changes at 705 nm were missing in TSSMs, suggesting this sample was devoid of PSI activity.

We then measured the kinetic of electrochromic shift (ECS) on leaves as a measure of protons transfer in the photosynthetic apparatus (Figure 10D). Briefly, the light-driven movement of electrons and protons through the thylakoid membranes generates a proton motive force (pmf) which comprises an electric field (ΔΨ) and a proton concentration gradient (ΔpH). The electric field also gets a shift of the absorption maxima of pigments embedded in a lipid membrane, a phenomenon called electrochromism. Such a “bandshift effect” can be detected at 520 nm to obtain the spectrum of the “field-induced absorption change,” namely the ECS signal, which shows a linear response to pmf. Upon steady-state actinic illumination of leaves, light-dark transition results in ECS decay, whose amplitude measures the light-driven pmf across the thylakoid membranes. TSSMs lacked detectable ECS signal, thus ruling out the generation of significant pmf in the chloroplasts of mites.

These results indicate that TSSM does not retain any photosynthetic activity despite the presence of intact chloroplasts and chlorophyll.

### 2.10. TSSMs Contain a Low Amount of Ribulose.1,5-Bisphosphate Carboxylase Oxygenase (RubisCO) Protein and Lack RubisCO Enzyme Activity

Another way to assess whether chloroplast contained in the TSSMs may contribute to support sugars to the TSSMs metabolism is by assessing the presence and activity of RubisCO. Total proteins extracted from healthy and TSSM-infested leaves, as well as from TSSMs, were analyzed by capillary gel electrophoresis in SDS conditions. Both RubisCO bands corresponding to the large, chloroplast-encoded subunit (56 kDA) and the small, nuclear-encoded subunit (14 kDa) are evident in both Lima bean healthy (Figure 11, H) and TSSM-infested (Figure 11, F) leaves, but are almost absent in TSSMs (Figure 11, SM). A 26 kDa band, most likely the light-harvesting chlorophyll-binding protein (LHCP) [21], is present in all samples, whereas an unidentified band at 46 kDa was only found in TSSMs (Figure 11).

The semi-quantitative analysis of the different polypeptides is reported in Table 3. In general, a significant (*p* < 0.05) and progressive reduction in the identified polypeptides was observed from healthy, TSSM-infested leaves to TSSMs, with the sole exception of a 37 kDa polypeptide that was significantly increased in TSSM-infested leaves and for a 46 kDa polypeptide that was significantly (*p* < 0.05) higher in TSSMs (Table 3). TSSM-infested leaves also showed a low-molecular-weight larger band corresponding to polypeptides (from 3 to 5 kDa) with respect to healthy leaves (Figure 11). The total content of RubisCO showed a significant (*p* < 0.05) reduction in TSSM-infested leaves with respect to healthy leaves, whereas in the TSSMs, only the large subunit of RubisCO was detectable (Table 3, Appendix A).

We then calculated the RubisCO activity in order to evaluate if the reduced content of the protein was associated with reduced enzyme activity. Healthy leaves had an enzyme activity of 46.24 ± 3.16 nmol min^−1^ mg^−1^ protein that was significantly (*p* < 0.05) higher than in TSSM-infested leaves (22.42 ± 2.02 nmol min^−1^ mg^−1^ protein). As expected, TSSMs showed a very low enzyme activity of RubisCO (9.17 ± 0.35 nmol min^−1^ mg^−1^ protein) that was significantly (*p* < 0.05) lower with respect to both healthy and TSSM-infested leaves.

## 3. Discussion

The biology of TSSMs is complex as the interaction with their host plants. TSSM feeding causes physiological and metabolic changes in the host plants that lead to a reduction in photosynthesis and, eventually, leaf senescence and death [2,22,23]. We provided new insights into the production of guanine and the catabolism of both Chl, carotenoids, and Pheo.

Previous studies demonstrated that crystals of synthetic guanine showed an autofluorescence identical to that observed inside the spider mites and their excreta [5]. In agreement with early studies [9], our results confirm the TSSMs synthesis and excretion of guanine. To our knowledge, this is the first report on the transcriptomics of the guanine pathway in TSSMs. Among the expressed genes, *PPAT*, *PAICS*, *ATIC,* and *GMPS* were the most upregulated (Figure 5). *PPAT* codes for an enzyme catalyzing the first step of purine nucleotide biosynthesis, whereas *PAICS* catalyzes the production of SAICAR, an intermediate metabolite in the de novo purine biosynthetic pathway [24]. *ATIC* encodes a bifunctional enzyme responsible for the catalysis of the last two steps in the de novo purine pathway [25], whereas *GMPS* encodes a glutamine amidotransferase that catalyzes the amination of the nucleotide precursor XMP to form GMP [26]. With regard to *GMPS*, several inhibitors of GMPS have been described (including 2-Iodo-ATP, glutamine, *N*-2-hydroxyguanosine 5’-monophosphate, ECC1385, etc.) [26,27,28,29] and DCY. The latter is a selective but reversible inhibitor of GMPS inhibiting the enzyme reversibly with uncompetitive inhibition kinetics toward glutamine and XMP and noncompetitive kinetics toward ATP [28]. DCY, which is an adenosine analog, is an unusual compound having a six-carbon ketose sugar, β-d-psicofuranosyl, in place of the more usual ribofuranose motif [30]. We found that DCY significantly reduced the feeding activity of TSSMs and these preliminary results suggest the potential of DCY as an agent of control for TSSMs infestations. Considering the constant threat to numerous crop plants due to TSSMs [31] and the continuous search for new and effective acaricides [32,33,34], we believe that DCY could be a good starting molecule to target guanine metabolism in the management of TSSMs infestations.

A clear effect of TSSMs feeding was found on carotenoids and Chls of Lima bean leaves. Carotenoids, a major group of tetraterpenic pigments, are known to support photosynthesis and provide protection against photo-oxidation. While it was presumed that all carotenoids in animals were sequestered from their diet, phylogenetic analyses suggest that in TSSMs, carotenoid biosynthesis was transferred from fungi into their genome, probably in a similar horizontal transfer [35] as suggested for aphids [36]. In TSSMs, carotenoids are responsible for the red-orange body colour and red eye colour [37,38,39]. Differences in the TSSM’s carotenoid biosynthesis gene expressions influence the abundance of carotenoid accumulation [40], indicating their role in body colour development [41]. Despite their ability to synthesize carotenoids, TSSMs use and contain the same carotenoids as host plants [35]. For instance, leaves of bean plants (*Phaseolus vulgaris*), which belong to the same genus of Lima bean, contain α-carotene, β-carotene, cryptoxanthin, lutein, lutein-5,6-epoxide, violaxanthin and neoxanthin [42], and the same compounds were found in TSSMs feeding on bean plants, with the addition of some ketocarotenoids such as astaxanthin. Similar changes in the carotenoid composition have been described in other mites such as *T. cinnabarinus* [42] and *T. pacificus* [43]. We showed that the total content of carotenoids of TSSMs exceeds about four times the Lima bean leaf carotenoid content on a weight basis and that TSSMs feeding activity lowers the leaf carotenoid content in infested tissues (Table 2).

In addition to reducing carotenoids, leaves infested by TSSMs showed altered and reduced leaf Chl content and composition [44,45]. We noticed that TSSM-infested leaves reduced their content of both Chl a and Chl b and increased the Pheo a and Pheo b concentration with respect to healthy leaves. We also noticed that the Chl a/Chl b ratio was typical of C3 plants [46]. These leaf metabolic changes are possible due to the feeding activity of TSSMs that contain manly Pheo and Chl degradation products. In the TSSMs, portions of chloroplasts and cytoplasm were observed in the ventricle, which agrees with other studies [47]. However, no photosynthetic activity was observed both in terms of quantum biology and enzymatic activity, as shown by our results on PSI and PSII functionality and RubisCO activity. These data agree with the chemical composition of TSSMs, which contain mainly Chl degradation products, with particular reference to Pheo a and Pheo b (and their epimers), that do not allow the functionality of the photosynthetic apparatus. Moreover, the low activity and presence of RubisCO, which is the key enzyme for CO_2_ carboxylation [48], further justifies the absence of any photosynthetic capacity of TSSMs.

The TSSM feeding activity increased the total sugar and total protein contents of infested Lima bean leaves. Plant protein and digestible carbohydrates are important nutritional resources for a number of herbivores, and the increased amount of these dietary components in infested leaves can be considered part of the strategy of TSSMs to increase fitness and improve both reproduction and mass movements [49]. However, our capillary gel electrophoresis analyses show a general reduction in RubisCO in TSSM-infested Lima bean leaves, which is paralleled by reduced enzyme activity and the appearance of a large band of low-molecular-weight polypeptides. The latter is probably the result of TSSM-induced protein degradation and agrees with the increased protein content in fed leaves. Indeed, the feeding activity of TSSMs has also been shown to cause a systemic increase in protein contents of young and uninfested leaves, with the production of specific proteins [44].

## 4. Materials and Methods

### 4.1. Mite Strains and Rearing

Two-spotted spider mites, *Tetranychus urticae* Koch 1836 (Acari: Tetranychidae), were reared on Lima bean (*Phaseolus lunatus* L.) plants for more than 10 years. These plants were grown in a greenhouse at a temperature of 22 ± 3 °C, relative humidity of 60–80%, and a photoperiod of L16:D8 h.

### 4.2. Epifluorescence Microscopy

TSSM-infested Lima bean leaf sections of about 100 mm^2^ were placed on a glass slide. A drop of 2-(N-morpholino)ethanesulfonic acid (MES) buffer (50 mM, pH 6.5) (Fluka) was placed on the leaf and covered with a glass coverslip. An additional buffer was gently perfused in order to get rid of air bubbles.

Slides were placed on the slide holder of a Nikon Eclipse 90i epifluorescence microscope, and protonymph stages of spider mites were observed by using bright field light (BF) and UV light. In the latter case, a Nikon UV-2A filter was used (Excitation 325–375 nm; Emission 420–800 nm). Photographs were taken using a Nikon DS-Fi1C Peltier-cooled 3CCD camera (Tokyo, Japan).

### 4.3. Confocal Laser Scanning Microscopy (CLSM)

TSSM-infested Lima bean leaf sections of about 100 mm^2^ were placed on a glass slide. A drop of MES buffer (50 mM, pH 6.5) was placed on the leaf and covered with a glass coverslip. An additional buffer was gently perfused in order to get rid of air bubbles. Mites were then observed with a Nikon D-Eclipse C1 spectral confocal laser scanner microscope (Tokyo, Japan). Chlorophyll and fluorescent chlorophyll degradation products were observed with a He-Ne laser by using an excitation wavelength of 637 nm and an LP of 650 nm. A red false color was used to indicate the chlorophyll fluorescence. Guanine was observed with an argon neon laser by using an excitation wavelength of 488 nm and an emission wavelength of 590 ± 50 nm, as previously described [12]. A green false color was used to indicate guanine. Images generated by the Nikon EZ-C1 3.80 software were elaborated by using the layers option of Adobe Photoshop^®^.

### 4.4. Transmission Electron Microscopy

Male and female TSSM (about 20 individuals per gender) were fixed by immersion in 3% (*v*/*v*) glutaraldehyde in a 100 mM sodium cacodylate buffer, pH 7.5, for 3 h under vacuum at room temperature. Samples were then washed in a sodium cacodylate buffer for 1 h and then pot-fixed in a 2% solution of osmium tetroxide in 50 mM sodium cacodylate buffer for 1 h at room temperature. Samples were then dehydrated with a graded series (30%, 50%, 70%, 95%, 100%) ethanol for 10 min for each step. 100% ethanol was repeated three times. Samples were then transferred to propylene oxide and used as a transitional fluid before resin embedding. Samples were then embedded in an Epon-Araldite concentration of propylene oxide-resin mixture and then polymerized for 24 h at 60 °C in pure Epon/Araldite resin. Thin sections were then cut with a Reichert Ultracut ultramicrotome. Thin sections were stained with lead citrate and uranyl acetate mounted on 400 mesh grids, and observed under the Philips CM10 transmission electron microscope (Amsterdam, The Netherlands).

### 4.5. RNA Preparation, cDNA Cloning, and qRT-PCR Assays

Total RNA was isolated and purified from three independent sampling of TSSMs (using a total of 1050 TSSMs) by using the Qiagen RNeasy Mini Kit and treating with RNase-Free DNase Set (Qiagen, Hilden, Germany) to remove genomic DNA. Sample quality and quantity were checked by using an Implen NanoPhotometer UV/Vis spectrophotometer (Implen GmbH, München, Germany) according to the manufacturer’s instructions.

One µg of the obtained total RNA was retro-transcribed into cDNA with random primers using the High-Capacity cDNA Reverse Transcription Kit (Applied Biosystems, Foster City, CA, USA), according to the manufacturer’s recommendations.

The obtained cDNA was diluted 1:10 and used for qRT-PCR assays. All experiments were performed on a Stratagene Mx3000P Real-Time System (La Jolla, CA, USA) using SYBR green I with ROX as an internal loading standard. The reaction was performed with a 25 µL mixture consisting of 12.5 µL 2X MaximaTM SYBR Green qPCR Master Mix (Fermentas International, Inc, Burlington, ON, Canada), 0.6 µL cDNA and 100 nM primers (Integrated DNA Technologies, Coralville, IA, USA). Controls included non-template controls (water template). PCR conditions were the following for all primers: 10 min at 95 °C, 45 cycles of 30 s at 95 °C, 30 s at 58 °C, and 20 s at 72 °C. Fluorescence was read following each annealing and extension phase. All runs were followed by a melting curve analysis from 55 to 95 °C. All amplification plots were analyzed with the MX3000PTM software to obtain Ct values. *Histone H3* was used as a reference gene.

Ct values were analyzed using the Δ^Ct^ method, and statistical differences between adults and nymphs were evaluated. Primers used for real-time PCR were designed using the Primer-BLAST (http://blast.ncbi.nlm.nih.gov/Blast.cgi accessed on October 2022) software and are shown in Appendix A.

### 4.6. Acaricidal Activity of Decoyinine (DCY)

One Lima bean (*Phaseolus lunatus* L.) segment of 400 mm^2^ were obtained from fully expanded leaves (cotyledonary leaves) and placed in a Petri dish containing a cotton layer moisturized with 0.5 mM 2-morpholinoethanesulfonic acid monohydrate (MES) buffer. At time 0, 0.5 mM, and 1 mM DCY dissolved in DMSO, and water were nebulized (1 mL) on different Petri dishes (at least 5 for each treatment) containing the Lima bean leaf segments. Five control plates were nebulized with 1 mL DMSO and water.

Ten adult female TSSM was then placed in both DCY and control nebulized plates. We estimated the effect of the DCY application by measuring the leaf damage inferred by the TSSM feeding activity after 4 and 7 days from the DCY application. The damage was assessed by image analysis using the ImageJ 1.52a software. Values are expressed as both the damages area (in mm^2^) and the percentage of leaf damage inferred by the feeding activity of TSSM. The number of replicates was always five.

### 4.7. Chlorophyll and Carotenoid Extraction

Chlorophylls, Chl degradation products, and carotenoids were extracted from healthy and TSSM-infested Lima bean leaves and from TSSMs according to the previously described protocol [20] with minor modifications. Briefly, 100 mg of healthy leaves, 100 mg of TSSM-infested leaves, or a number of both adult and young (female and male) TSSMs corresponding to 100 mg were weighed and ground in liquid nitrogen. To each sample, 1 mL of acetone was added to extract all pigment compounds. After vortexing for 30 s and centrifugation (5000× *g*, 5 min, 4 °C), the supernatant was separated from the solid residue and stored in a clean glass tube. During sample extraction, a gentle sonication (2 min, RT) was applied to increase the extraction efficiency. The extraction step was repeated 5 more times, and the various acetone phases were collected together. After pooling, the acetone fraction was dried under nitrogen flow and then dissolved in 6 mL of dimethylformamide (DMF). The DMF fraction was treated with 2 mL of hexane and centrifuged at 2000 g. Hexane extraction was repeated two more times, and the three hexane phases were collected together. The hexane layer was separated and stored in a new collection tube. Chlorophylls and Chl degradation products were retained in the DMF phase, while the hexane phase contained carotenoids. The DMF phase was treated with 6 mL 2% (w/v) aqueous solution of Na_2_SO_4_ in an ice bath, followed by three serial additions of 1 mL hexane:ethyl ether in a 1:1 (*v*/*v*) ratio. The aqueous phase was discarded, allowing the removal of polyphenols and other interfering water-soluble compounds. The organic phase was reconstituted to a known volume, aliquoted, and completely dried under nitrogen flow. The dry residue was dissolved in a suitable volume of a mixture consisting of methanol (MetOH) and methyl tertiary butyl ether (MTBE) in a 1:1 (*v*/*v*) ratio. Before injection into HPLC, both the Carotenoid- and Chl-containing portions were centrifuged at 10,000 *g*, at 4 °C, for 10 min.

### 4.8. Liquid Chromatography of Chlorophyll, Chlorophyll Degradation Products, and Carotenoids

Chlorophylls, Chl degradation products, and CARs extracted from samples were analyzed with a high-performance liquid chromatography (1200 HPLC, Agilent Technologies, Santa Clara, CA, USA). Chromatographic separation, identification, and quantifications of the compounds were performed as previously described [20]. Briefly, in a thermally (25 °C) equilibrated C30 column (250 mm × 2.1 mm i.d., 3 μm, YMC America, Devens, MA, USA), the mobile phase consisting of 95% (*v*/*v*) Solvent A [90% (*v*/*v*/*v*) MetOH, 3% (*v*/*v*/*v*) MTBE, and 5% (*v*/*v*/*v*) H_2_O] and 5% (*v*/*v*) Solvent B [88% (*v*/*v*/*v*) MTBE, 10% (*v*/*v*/*v*) MetOH, and 2% (*v*/*v*/*v*) H_2_O] was fluxed. To both solution A and solution B, 50 mM Ammonium Acetate was added. In order to separate the pigments of interest, the various proportions of Solvent A and Solvent B shown in Appendix A were flushed at a constant flow rate of 0.2 mL/min. Chromatograms were integrated for quantification at the following wavelengths: 661 nm (chlorophyll a and a′), 642 nm (chlorophyll b and b′), 667 nm (pheophytin a and a′), 655 nm (pheophytin b and b′) and 460 nm (carotenoids). Pigment identification was according to the literature data [50] and injection of pure standards.

### 4.9. Liquid Chromatography and Tandem Mass Spectrometry and Guanine

Guanine quantification was obtained HPLC (see above) coupled to a mass spectrometer (6330 Series Ion Trap Mass Spectrometer System, Agilent Technologies, Santa Clara, CA, USA) provided with an electrospray ionization (ESI) module. ZORBAX Eclipse XDB-C18 columns (150 mm × 4.6 mm i.d., 3.5 μm) were used for separation. The solvents for the mobile phase were milliQ water (solvent A) and methanol (solvent B). In both eluent solvents, 5 mM ammonium acetate was added. The gradient used for elution consisted of a linear gradient in which solvent B rose from 0% (*v*/*v*) to 5% (*v*/*v*) in 10 min, then to 20% (*v*/*v*) in 20 min, and finally at 100% (*v*/*v*) solvent B in10 min. The column was reconditioned by using 0% (*v*/*v*) of solvent B, and 100% (*v*/*v*) of Solvent B for 5 min. The flow rate was set at 0.2 mL min^−1^, and the column was thermostated at 25 °C. Peaks were detected at 254 nm UV detection and in positive ion mode for MS and MS/MS detection. Mass spectrometry was performed in scan mode from m/z 50–350 uma. To select suitable precursor ions and product ions, ESI-MS/MS parameters, including fragmentation amplitude, compound stability, and product ions, were automatically optimized with the aim of obtaining the maximum signal for each analyte using a syringe pump directly connected to the ion trap mass spectrometer via an electrospray ionization interface (ESI). The fragmentation cut-off was set as default, and the isolation amplitude was 4. For ionization in ESI, the drying gas N_2_ was set at 10 L min^−1^, the temperature at 350 °C, nebulizer pressure at 40 p.s.i., and capillary voltage at 4.5 kV.

### 4.10. Photosynthetic Activity in TSSMs

TSSM was isolated from Lima bean-infested cotyledonary and primary leaves by vortexing 3 × 30 s. in the presence of 50 mM PBS buffer using 50 mL Falcon tubes. Leaves were then removed, and the tubes were centrifuged at 5000 rpm in order to obtain pelleted TSSMs. The supernatant was then removed, and TSSMs were weighted, isolated, frozen in liquid nitrogen, and kept at −80 °C.

In order to assess the photosynthetic activity of TSSMs, we first determined the 77K fluorescence emission spectra of chloroplasts from Lima bean leaves and of TSSM extracts. This analysis can distinguish among the typical emission bands which originate from Chl-binding complexes associated with each photosystem [51]. Chlorophyll fluorescence emission spectra of leaf chloroplasts and TSSM extracts were then recorded in liquid nitrogen (77 K). Chloroplasts were prepared from Lima bean leaves and homogenized using mortar and pestle in GB buffer [52], while TSSMs were ground in 1.5 mL tube and pestle with 1 mL GB. Both samples were diluted 1:4 in 10 mM Hepes pH 7.5, 50% w/v glycerol, and 1 mL of each suspension was transferred in NMR capillary tubes (Sigma-Aldrich Z272019, St. Louis, MI, USA) and immediately frozen in liquid nitrogen. Chlorophyll fluorescence emission spectra were recorded with a Jobin-Yvon Fluoromax-3 spectrofluorimeter. Excitation light was at 440 nm (10 nm slit width), and emission was detected from 660 to 780 nm (5 nm slit width). All spectra were normalized to the max peak.

PSII function was measured in vivo through light-induced chlorophyll fluorescence kinetics. 0JIP transient [53] from minimum (F_0_) to maximum (F_MAX_) values were measured at room temperature (22 °C) with a home-built setup. Samples (leaf discs or TSSMs immobilized on transparent tape) were excited with green actinic light (570 nm, 1100 µmol photons m^−2^ s^−1^ for 1 s), and emission was detected in the near far-red [54]. All samples were dark-adapted for 15 min before the fluorescence measurements were carried out. PSI function was measured in vivo through absorption changes at 705 nm, which reflect the events of charge separations in the PSI reaction center P700. Measurements were performed on Lima bean leaves and TSSMs, using a LED spectrophotometer (JTS10, Biologic Science Instruments, Seyssinet-Pariset, France). Absorption changes were sampled by weak monochromatic flashes (705 nm, 10-nm bandwidth), while charge separation within PSI was induced by far-red actinic light (730 nm, 180 µmol photons m^−2^ s^−1^ for 30 s, followed by 8 s of dark relaxation). The amplitude of light-induced trans-thylakoid pmf (protonmotive force) was estimated in Lima bean leaves and TSSMs from changes in absorbance associated with the ECS (electrochromic shift) at 520 nm [55] in a JTS10. Absorption changes were sampled by weak monochromatic flashes (520 nm, 10–nm bandwidth) while ECS was induced by actinic light (640 nm, 940 µmol photons m^−2^ s^−1^ for 120 s, followed by 20 s of dark relaxation).

### 4.11. Total Sugar Content of Lima Bean Leaves and TSSMs

The total carbohydrates from healthy and TSSM-infested *P. lunatus* leaves were measured using the Total Carbohydrate Assay Kit (MAK104, Sigma-Aldrich, St. Louis, MI, USA) according to the manufacturer’s instructions. 50 mg of healthy and TSSM-infested leaves were homogenized in 200 µL ice-cold Assay Buffer. Samples were then centrifugated at 13,000 *g* for 5 min, and the supernatant was used for the assay. Samples (30 µL) were loaded to a flat bottom 96 well plate with the addition of 150 µL sulfuric acid. After 15 min, 30 µL of Developer was added to each well at 90 °C in the dark. The absorbance was read at 490 nm with a Microplate Reader (NB-12-0035, Neo Biotech, Nanterre, France). The number of total carbohydrates present in samples was determined from a standard curve made by dilutions of 2 mg mL^−1^ d-glucose standard.

### 4.12. Total Protein Content of Lima Bean Leaves and TSSMs

The total protein content was obtained using an extraction buffer made of 62.5 mM Tris-HCl pH7, 10% Glycerol, 2% SDS, 1 mM EDTA, and 1 mM PMSF. One hundred mg leaves and 40 mg TSSMs were ground in liquid nitrogen. Samples were heated at 85 °C for 10 min and then centrifuged at 18,000 *g* for 5 min at room temperature (RT). The supernatant was then precipitated with 1:4 (*v*/*v*) 100% acetone, kept at −20 °C for 3 h, and then centrifuged at 18,000 *g* for 7 min at RT. The supernatant was discarded, and the pellet was dry on air dry for 20 min. Pellets were resuspended with a solution of 5 mM Tris-HCl pH 7, 1 mM EDTA, and 0.1% SDS. Samples were then used for the protein electrophoresis and quantified by Coomassie (Bradford, UK) Protein Assay Kit (Thermo Fisher Scientific, Waltham, MA, USA).

### 4.13. Capillary Gel Electrophoresis

Lima bean and TSSMs proteins were characterized with the Agilent Protein 80 Kit by using the Agilent 2100 Bioanalyzer (Agilent Technologies, Santa Clara, CA, USA) according to the manufacturer’s instructions. The same concentration of proteins for each sample obtained from the total protein extraction was used to prepare samples, reaching a final volume of 4 µL. The Protein 80 Chip was then analyzed in the 2100 Bioanalyzer, and the data acquisition and analysis were performed by the Agilent 2100 Expert Software (Agilent Technologies, Santa Clara, CA, USA).

### 4.14. RubisCO Extraction and Enzyme Activity

Enzyme extraction and purification were conducted at 4 °C. A ratio of 1:6 (w/v) cold extraction buffer (50 mM NaOH-Bicine pH 8.2, 20 mM MgCl_2_, 1 mM EDTA, 2 mM Benzamidine, 5 mM aminocaproic acid, 50 mM 2-mercaptoethanol, 10 mM DL-dithiothreitol (DTT), 1 mM phenylmethylsulfonyl fluoride (PMSF)) was used to extract healthy, TSSM-infested Lima bean leaves and TSSMs that were ground in liquid nitrogen was added. The homogenate was then centrifuged at 14,000 *g* for 1 min at 4 °C. The supernatant was transferred to a new ice-cold tube and immediately used for the RubisCO activity assay.

The assay was carried out in a Microplate reader (NB-12-0035 Neo Biotech, Nanterre, France). In flat bottom 96-well polystyrene microplates, 159.9 µL of mQ water for the blank and 153.9 µL for the samples were pipetted into each well, followed by 35.1 µL assay mix (100 mM NaOH-Bicine pH 8.2, 20 mM MgCl_2_, 10 mM NaHCO_3_, 20 mM KCl, 5 mM DTT, 2 UI 3-Phosphoglyceric phosphokinase, 0.4 UI α-Glycerophosphate Dehydrogenase, 24 UI Triosephosphate isomerase, 2.8 UI Glyceraldehyde 3-Phosphate Dehydrogenase, 3 mM ATP, 1 mM NADH) avoiding exposure to light. Six µL 20 mM RuBP was added to start RubisCO activity. Then, 5 µL of sample supernatant was added to the wells. The Microplate reader was set at 30 °C, and absorbance was read at 340 nm. The calculation of RubisCO activity was carried out according to Sales et al. [56].

### 4.15. Statistical Analysis

At least five biological repetitions were used. The data are expressed as mean values; metric bars indicate the standard deviation (SD). To evaluate the different significance of the adult and the nymphs, ANOVA was performed. Tukey test was also conducted by using the Bonferroni test (*p*-value of either < 0.05, < 0.01, or < 0.001) was used to determine post-hoc differences.

## 5. Conclusions

Resistance management strategies to fight TSSM infestations are based on the deep knowledge of biology and physiology of spider mites and how they affect the metabolism of infested plants. Here we provided new information on the ability of TSSMs to catabolize leaf saps and on the metabolism of guanine, the nitrogen waste product of these tetranychids. In infested leaves, TSSMs interaction with the host plant increases the total content of proteins and soluble sugars but reduces RubisCO content and activity. These events precede leaf senescence and, eventually, leaf death. The plant cell-sucking activity of TSSMs is associated with the presence of intact chloroplasts in the TSSMs midgut; however, no photosynthetic activity is present due to the lack of either functional PSII centers, PSI activity, or the generation of significant pmf in the mites chloroplasts. These results depended on the fact that most of the Chl-proteins detected in TSSM extracts are not part of functioning photosystems and are made mainly by pheophytins and other products of Chl catabolism. Furthermore, RubisCO is almost absent in TSSM, preventing any carboxylation reaction. With regards to the possible new targets for acaricidal activity, the transcriptomics of guanine biosynthesis identified *GMPS* as one of the most expressed genes and our preliminary results on DCY, a selective inhibitor of GMPS, show its potential for further studies aimed at reducing TSSMs infestations.

## Figures and Tables

**Figure 1 ijms-24-01715-f001:**
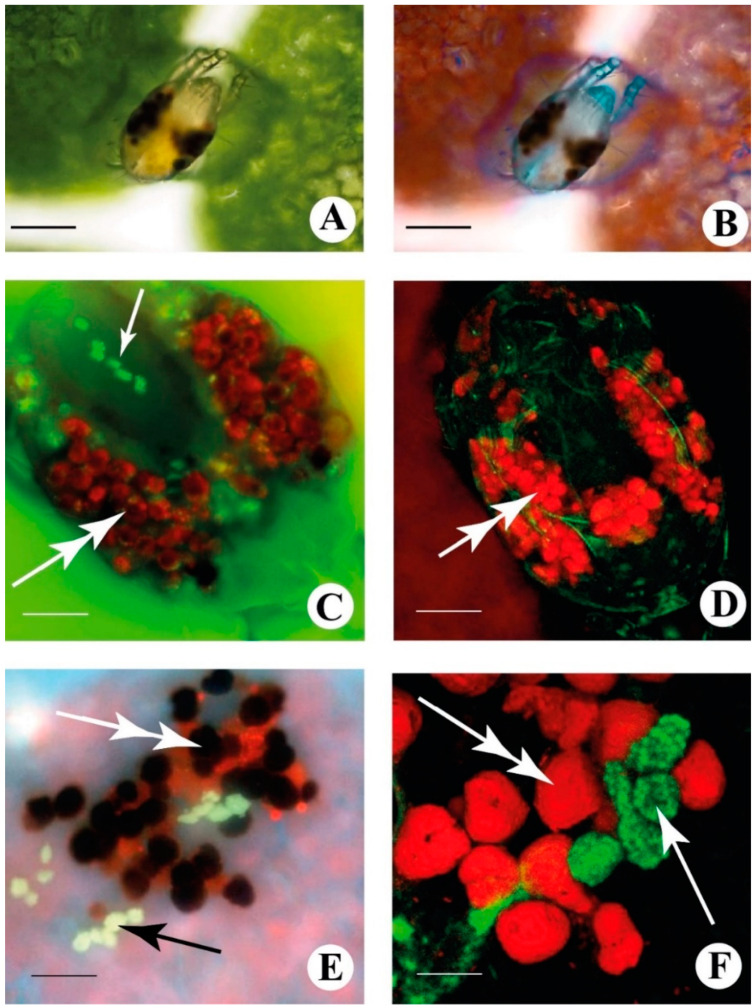
TSSMs and their excreta are used in this study. (**A**) TSSM observed in bright fields. (**B**) TSSM was observed with epifluorescence microscopy. (**C**) enlargement of an epifluorescence view of a TSSM that shows the presence of fluorescent guaninesomes (single arrow) in the excretory organ and the red fluorescence of chlorophyll inside the mite caeca (double arrow). (**D**) TSSM observed with CLSM. The double arrow indicates the chlorophyll fluorescence inside the lateral caeca. (**E**) epifluorescence of TSSM excreta (pellets) composed of chlorophyll degradation products showing different levels of red fluorescence (double arrow) and guaninesomes showing a bright fluorescence (single black arrow). (**F**) TSSM pellets observed using CLSM. Red fluorescence is associated with chlorophyll degradation products (double arrow), while guaninesomes are differentiated by green fluorescence (single arrow). Metric bars A, B = 80 µm; C-E = 30 µm; F = 24 µm.

**Figure 2 ijms-24-01715-f002:**
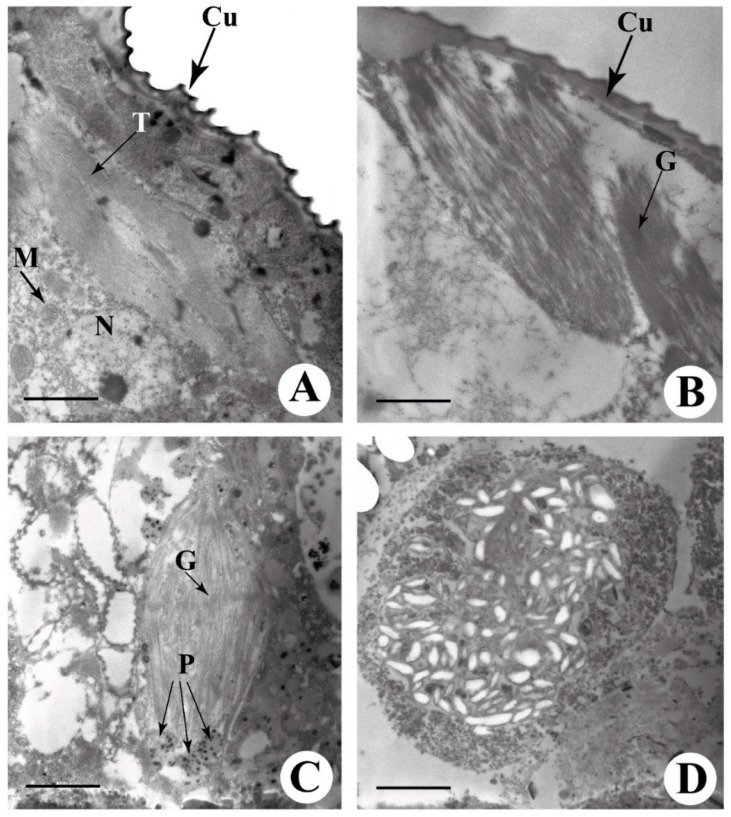
Transmission electron microscopy of TSSM. (**A**) ultrathin section of a TSSM showing the presence of the mite cuticle (Cu), a chloroplast showing the thylakoids (T), mitochondria (M), and a nucleus (N). (**B**) ultrathin section of a TSSM showing the presence of the mite cuticle (Cu) and the grana (G) stacks of a chloroplast. (**C**) ultrathin section of a TSSM showing a chloroplast with evident grana (G) stacks and the presence of some plastoglobules (P). (**D**) ultrathin section of a TSSM showing the ultrastructure of one of the red fluorescent bodies indicated by the double arrow in Figure 1C,D. Metric bars, A = 1.57 µm; B, C = 1.02 µm; D = 3.46 µm.

**Figure 3 ijms-24-01715-f003:**
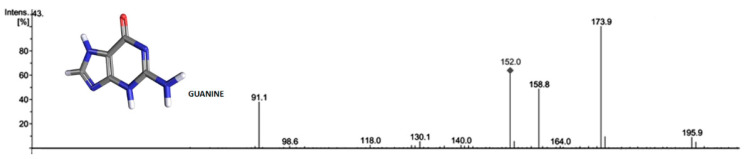
Mass spectrum of guanine identified in TSSMs and in their excreta.

**Figure 4 ijms-24-01715-f004:**
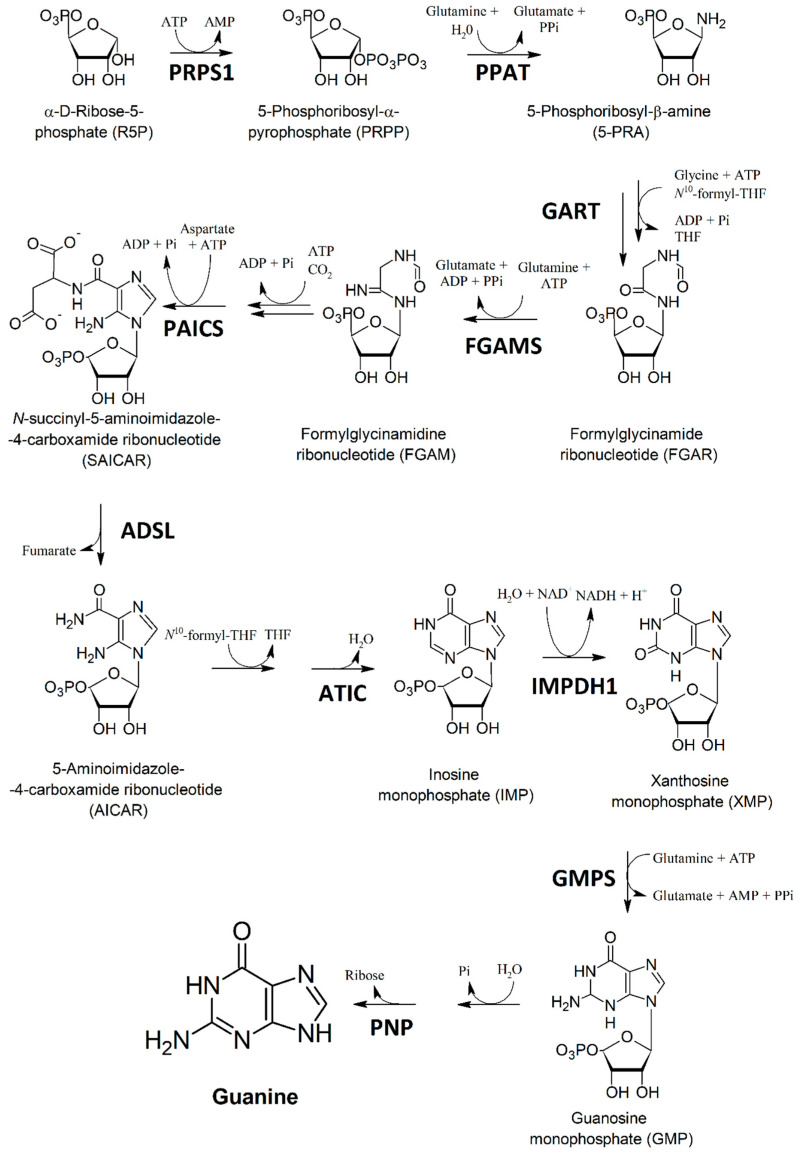
Biosynthetic pathway of guanine. (PRPS1) 5-phosphoribosyl-1-pyrophosphate synthetase; (PPAT) glutamine phosphoribosyl pyrophosphate aminotransferase; (GART) glycinamide ribonucleotide formyltransferase; (FGAMS) phosphoribosyl formyl glycinamidine synthase; (PAICS), phosphoribosylaminoimidazolesuccinocarboxamide synthase; (ADSL) adenylosuccinate lyase; (ATIC) AICAR transformylase/IMP cyclohydrolase; (IMPDH1) IMP dehydrogenase; (GMPS) GMP synthetase; (PNP) Purine nucleotide phosphorylase.

**Figure 5 ijms-24-01715-f005:**
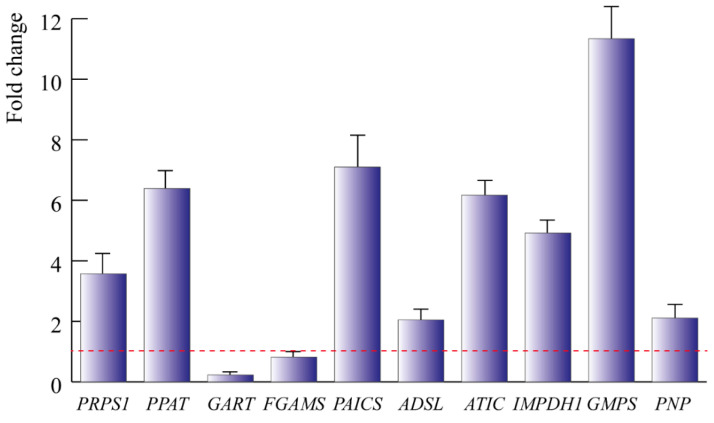
qPCR guanine gene expression of TSSM. Bars indicate the standard deviation over the mean of at least three biological replicates. The red dotted bar represents the reference gene expression; values above the dotted line indicate upregulation.

**Figure 6 ijms-24-01715-f006:**
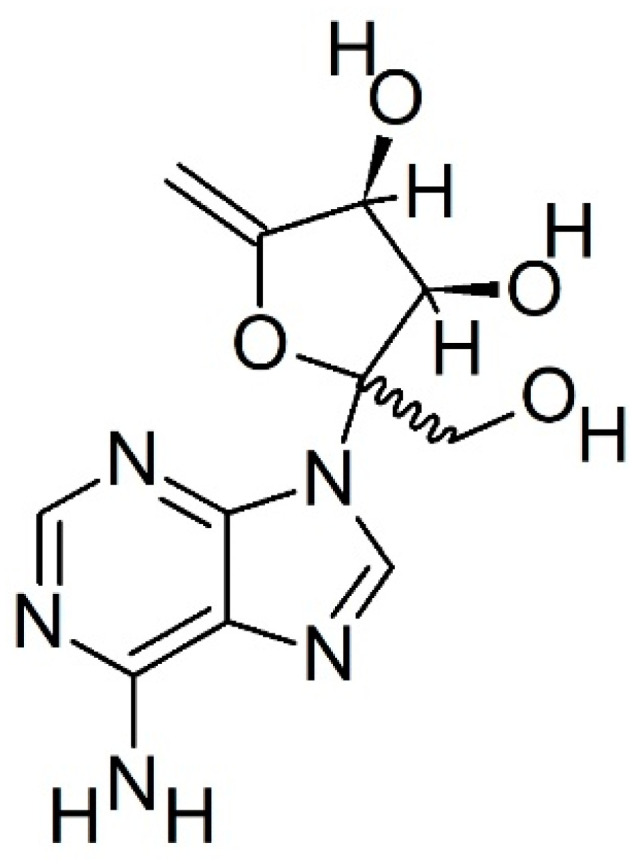
Chemical structure of decoyinine (DCY).

**Figure 7 ijms-24-01715-f007:**
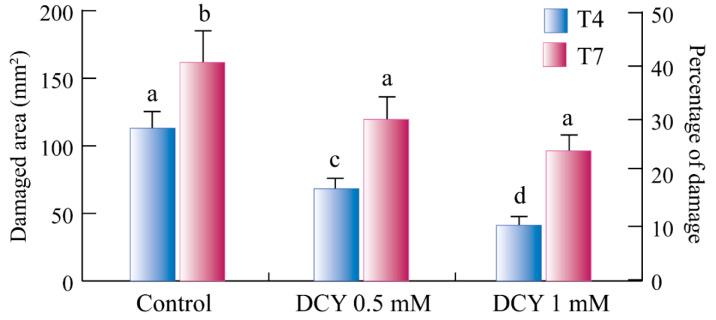
Effect of decoyinine (DCY) on TSSM. Bar graphs represent the amount of damage (left axis) and the percentage of damage (right axis) inflicted by TSSM on Lima bean (*Phaseolus lunatus*) leaves. Controls (without the use of DCY) and two concentrations of DCY (0.5 mM and 1 mM) were used to evaluate the amount of damage inflicted by TSSMs at 4 and 7 days (T4 and T7) after the onset of the experiment. Error bars indicate standard deviation; different letters indicate significant (*p* < 0.05) differences.

**Figure 8 ijms-24-01715-f008:**
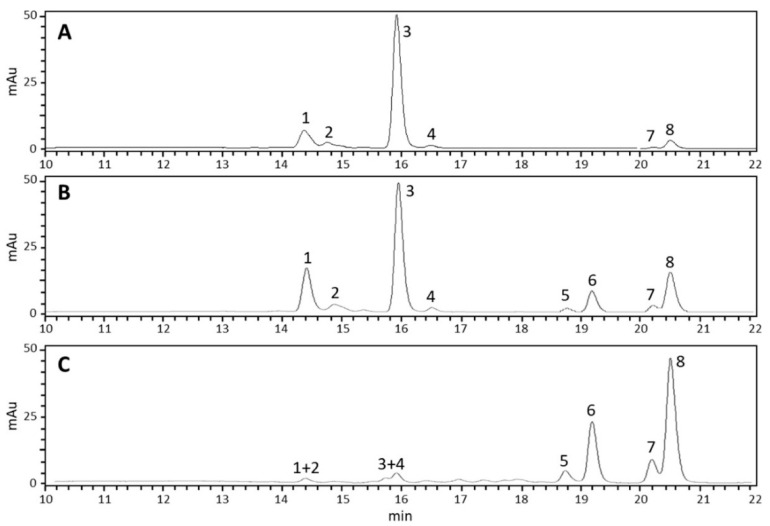
HPLC chromatograms (DAD λ655 nm) showing Chl and Chl degradation products in healthy Lima bean leaves (**A**), TSSM-infested Lima bean leaves (**B**), and TSSMs (**C**). Numbers refer to: Chl b (1); Chl b’ (2); Chl a (3); Chl a’ (4); Pheo b’ (5); Pheo b (6); Pheo a’ (7); Pheo a (8). UV/Vis spectra of the eight compounds are shown in Appendix A.

**Figure 9 ijms-24-01715-f009:**
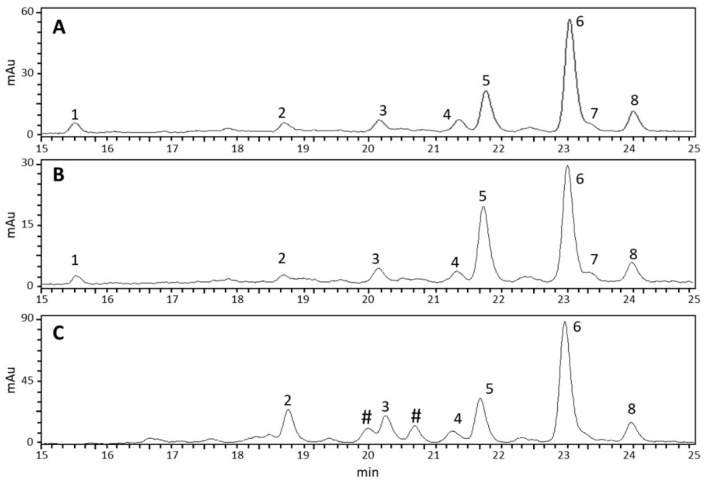
HPLC chromatograms (DAD λ 450 nm) showing carotenoids in healthy Lima bean leaves (**A**), TSSM-infested Lima bean leaves (**B**), and TSSMs (**C**). Numbers correspond to Lutein (1); unidentified carotenoid#1 (2); 15-cis-β-carotene (3); 13-cis-β-carotene (4); trans-α-carotene (5); trans-β-carotene (6); cis-α-carotene (7); 9-cis-β-carotene (8); γ-carotene (not shown). The symbol **#** indicates peaks resulting from Pheo a’ and Pheo contamination (see Figure 8C). UV/Vis spectra of the nine compounds are shown in Appendix A.

**Figure 10 ijms-24-01715-f010:**
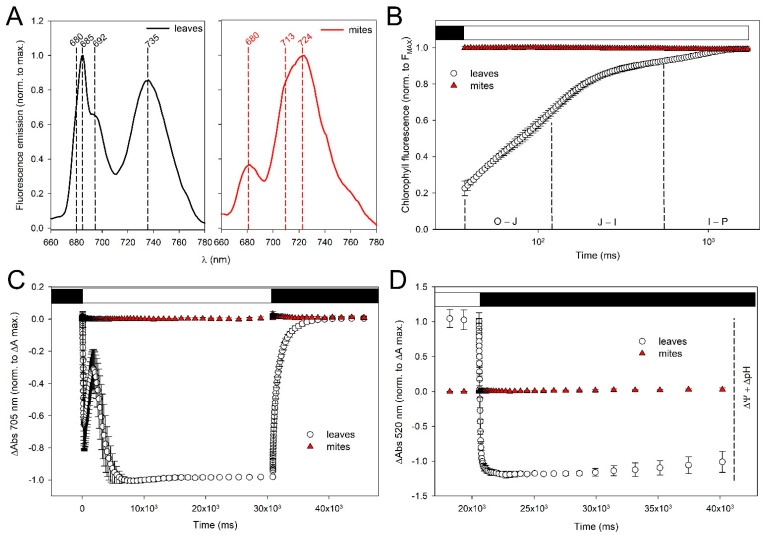
Photosynthetic activity in Lima bean leaves and in the TSSM. (**A**) 77K fluorescence emission spectra were recorded on chloroplasts from Lima bean leaves and on TSSMs. λ_exc_ = 440 nm, λ_emiss_ = 660–780 nm. Each spectrum was normalized to the maximum emission. Data are expressed as mean values of three fluorescence curves. (**B**) PSII fluorescence induction kinetics (0JIP transient). Fluorescence rise was induced on dark-adapted leaves and TSSMs, using a saturating flash of green light (1100 µmol photons m^−2^ s^−1^ for 1 s). Traces have been normalized to the F_MAX_ value. (**C**) P700 oxidation kinetics. Absorbance changes at 705 nm were measured in Lima bean leaves and TSSMs and normalized to the maximum ΔAbs measured in leaves. The samples were illuminated with far-red actinic light (730 nm, 1500 µmol photons m^−2^ s^−1^ for 15 s) followed by 15 s of dark relaxation. (**D**) Kinetics of the ECS signal normalized to the maximum ΔAbs measured in leaves at the end of the light window. The samples were illuminated with actinic light (640 nm, 940 µmol photons m^−2^ s^−1^ for 120 s) followed by 20 sec of dark relaxation. The transient changes of the ECS, upon switching the light off, reflect the different rates of relaxation of the two components of the transmembrane protonmotive force ΔΨ and ΔpH. In panels (**B**–**D**), symbols and error bars show means ± SD n = 4, and black and white bars represent dark and light phases, respectively.

**Figure 11 ijms-24-01715-f011:**
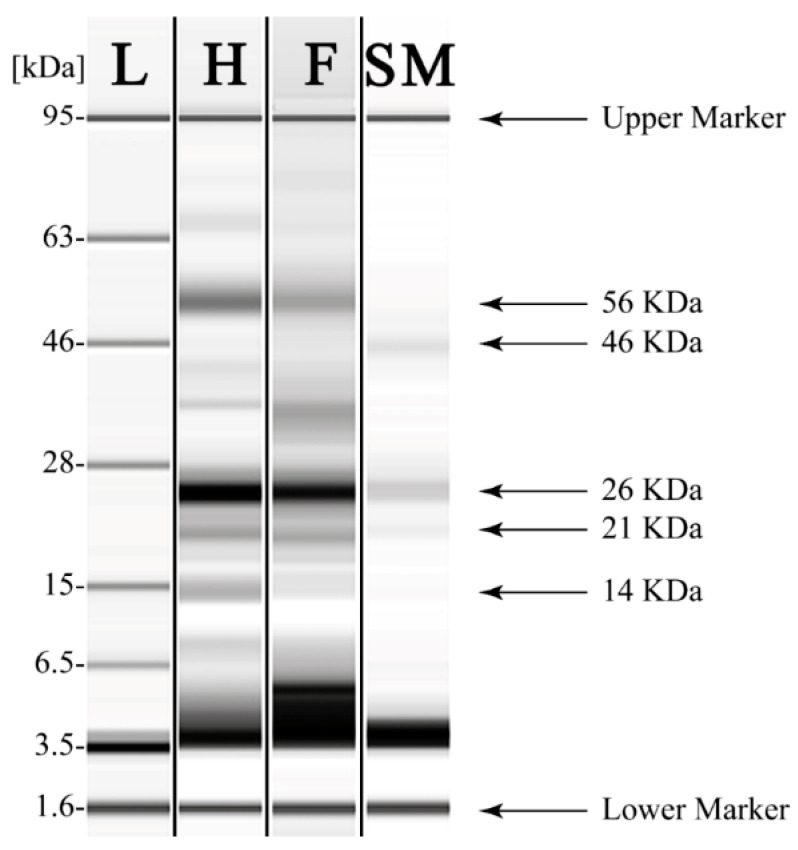
SDS capillary gel electrophoresis (Agilent Bioanalyzer) of proteins extracted from Lima bean healthy (H) leaves, TSSM-infested (F) Lima bean leaves, and TSSM (SM). Two clear bands of RubisCO (at 14 and 56 kDa) are evident in the leaves but almost absent in TSSMs. L, ladder.

**Table 1 ijms-24-01715-t001:** Chemical composition of chlorophylls and chlorophyll degradation products in healthy and TSSM-infested Lima bean leaves and in TSSMs. Values are reported as mean (± standard deviation) values of at least three biological replicates. For each raw, different letters indicate significant differences (*p* ≤ 0.05). For further statistical information, see Appendix A.

Compound	RT (min)	Healthy Leaf	TSSM-Infested Leaf	TSSM
Chl b	14.6	20.603 (0.619) ^a^	15.323 (0.511) ^b^	0.92 (0.034) ^c^
Chl b’	15.1	1.298 (0.018) ^b^	0.661 (0.014) ^a^	0.622 (0.022) ^a^
Chl a	16.3	77.444 (1.472) ^a^	62.585 (2.447) ^b^	1.555 (0.048) ^c^
Chl a’	16.9	15.185 (0.457) ^a^	9.789 (0.392) ^b^	1.963 (0.021) ^c^
Chl a/b ratio		3.75	4.08	1.68
Pheo b’	19.2	n.d.	1.005 (0.032) ^b^	5.809 (0.211) ^a^
Pheo b	19.5	n.d.	0.652 (0.024) ^b^	1.21 (0.017) ^a^
Pheo a’	20.3	0.594 (0.009) ^b^	1.49 (0.016) ^b^	38.925 (1.188) ^a^
Pheo a	20.3	0.069 (0.002) ^a^	0.21 (0.004) ^b^	3.649 (0.083) ^c^
TChP		115.193 (2.577) ^a^	91.715 (3.44) ^b^	54.653 (1.624) ^c^

RT: retention time; TChP: Total Chl and Pheo content; n.d.: not detected.

**Table 2 ijms-24-01715-t002:** Quantification of carotenoids in Lima bean healthy and TSSM-infested leaves and in TSSM. Values are reported as mean (±standard deviation) values of at least three biological replicates. For each raw, different letters indicate significant differences (*p* ≤ 0.05). For further statistical information, see Appendix A.

Compound	RT (min)	Healthy Leaf	TSSM-Infested Leaf	TSSM
Lutein	15.3	5.602 (0.014) ^a^	1.645 (0.034) ^b^	n.d.
Putative xanthophyll	18.6	5.154 (0.15) ^b^	5.464 (0.188) ^b^	44.545 (0.467) ^a^
15-cis-β-carotene	20.8	2.822 (0.052) ^b^	2.108 (0.046) ^c^	22.588 (0.372) ^a^
13-cis-β-carotene	21.5	8.646 (0.187) ^b^	6.671 (0.197) ^c^	48.061 (0.531) ^a^
trans-α-carotene	21.8	15.57 (0.167) ^b^	10.661 (0.263) ^c^	19.926 (0.502) ^a^
cis-α-carotene	22.6	1.192 (0.034) ^c^	8.006 (0.123) ^b^	9.723 (0.346) ^a^
trans-β-carotene	23.0	153.664 (3.95) ^b^	121.473 (4.561) ^c^	630.804 (24.929) ^a^
9-cis-β-carotene	23.5	19.331 (0.632) ^a^	15.83 (0.364) ^b^	n.d.
γ-carotene	24.2	5.273 (0.019) ^b^	3.981 (0.253) ^c^	65.127 (0.595) ^a^
TCrC		222.453 (5.332) ^b^	179.286 (6.141) ^c^	849.482 (28.071) ^a^

TCrC: Total content of carotenoids, n.d., not detectable.

**Table 3 ijms-24-01715-t003:** The relative concentration of polypeptides was calculated from the capillary gel electrophoresis of protein extracts from healthy and TSSM-infested Lima bean leaves and from TSSMs. Values are expressed as µg ml^−1^ (± standard deviation). In the same row, different letters indicate significant (*p* < 0.05) differences; n.c., not computable.

Polypeptide Size (kDa)	Healthy Leaves	TSSM-Infested Leaves	TSSM
14.0	218.2 (±3.9 ^a^)	34.2 (±2.2 ^b^)	n.c.
18.0	29.2 (±1.2 ^a^)	10.0 (±2.9 ^b^)	n.c.
21.0	243.5 (±28.0 ^a^)	128.0 (±11.1 ^b^)	30.1 (±4.5 ^c^)
22.0	82.4 (±3.6)	n.c.	n.c.
26.0	1224.5 (±144.0 ^a^)	747.9 (±32.0 ^b^)	179.9 (±14.4 ^c^)
37.0	51.9 (±5.7 ^a^)	262.7 (±13.3 ^b^)	n.c.
42.0	47.1 (±6.2 ^a^)	n.c.	8.3 (±2.9 ^b^)
46.0	4.1 (±1.0 ^a^)	6.8 (±3.0 ^a^)	63.5 (±5.4 ^b^)
56.0	364.5 (±21.2 ^a^)	179.6 (±11.6 ^b^)	3.6 (±0.5 ^c^)
Total content	2284.5 (±223.6 ^a^)	1319.0 (±117.1 ^b^)	279.2 (±18.0 ^c^)
RubisCO	582.7 (±24.1 ^a^)	202.4 (±18.8 ^b^)	3.6 (±0.5 ^c^)

## Data Availability

Data are available upon request.

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
