# Peer review of "Biology of Two-Spotted Spider Mite (Tetranychus urticae): Ultrastructure, Photosynthesis, Guanine Transcriptomics, Carotenoids and Chlorophylls Metabolism, and Decoyinine as a Potential Acaricide"

_ijms, 2023, doi:10.3390/ijms24021715_

Round 1

Reviewer 1 Report

Dear authors, I have reviewed the manuscript entitled "Biology of Two-Spotted Spider Mites (Tetranychus urticae): Ultrastructure, Photosynthesis, Guanine Transcriptomics, Ca-rotenoids and Chlorophylls Metabolism, and Decoyinine as a Potential Acaricide"

Within the document, you can see the suggestions made.

Author Response

We thank the reviewer for the careful revision of the manuscript that allowed us to implement and improve the text. Below are the point-to-point changes made in the revised text according to the reviewer’s comments.

Page 1 line 2, remove s from Mites

R: done

Page 1 line 16, Koch 1836 (Acari: Tetranychidae)) For being mentioned for the first time in the abstract

R: done

Page 1 line 35, Koch 1836 (Acari: Tetranychidae) For being mentioned for the first time in the main text

R: done

Page 4 line 125, remove s from TSSMs

R: done

Page 4 line 140, remove s from TSSMs

R: done

Page 5 lines 148-149: Rewrite the sentence. As it is written it belongs to the discussion section.

R: as suggested this sentence has been moved to the discussion section (page 14 lines 423-424)

Page 5 lines 150-151: As it is written, it belongs to the Materials and Methods section. As it is written, it belongs to the materials and methods section. Only include the results obtained

R: the sentence has been rephrased

Page 5 line 158, delete sentence

R: the sentence has been deleted and the next sentence rephrased

Page 7 line 195, remove s from TSSMs

R: done

Page 7 line 198, remove s from TSSMs

R: done

Page 7 line 200: Just write the results found. Switch to the discussion section.

R: the sentence has been rephrased.

Page 7 line 202, remove s from TSSMs

R: done

Page 8 line 213, remove s from TSSMs

R: done

Page 8 lines 220-228, Just write the results found. The wording of this paragraph is a combination of the results and materials and methods sections.

R: the sentences have been rephrased in order to avoid repetition with the Materials and Methods.

Page 8 line 230, replace with damaged by TSSM feeding

R: done

Page 8 line 232, Change it to the discussion section.

R: The sentence has been modified and the reference to C3 plants has been moved to discussion (page 15 lines 489-490)

Page 8 line 238, To rewrite. Fed or infested? Or In Lima bean leaves damaged by TSSM feeding, This sentence is found in several parts of the manuscript, check if it is  what you want to say in the sentence.

R: the term “fed” has been consistently replaced by the term “infested” throughout all the text

Page 10 line 283, TSSM-fed leaves?

R: the term “fed” has been consistently replaced by the term “infested” throughout all the text

Page 11 line 305, change increased with lower that and fed with infested

R: done

Page 11 lines 310-313, Move these lines into the materials and methods section. If it is already in that section, please remove it from here.

R: the sentence has been removed and moved to the Materials and Methods section 4.10 (pages 18 and 19 lines 691-694)

Page 12 line 354 remove s from TSSMs

R: done

Page 12 line 357 remove s from TSSMs

R: done

Page 14 line 404 remove s from TSSMs

R: done

Page 15, lines 452-453, Why? Clarify

R: the sentence has been rephrased to clarify the concept

Page 15 line 458, change fed with infested

R: done

Page 15 line 463, change with which agrees with other studies

R: done

Page 16 line 483, Remember that the reader will want to repeat the experiment, so you must be as precise in the methodology.

R: we thank the reviewer for reminding this important issue. In agreement with the Journal’s guidelines and in order not to overlap the text with other publications, we limited the text to the essential and referred to cited papers as a reference to the methods described.

Page 16 line 484, How many TSSM did you use for each trial? What generation were the spider mites? Were they males or females?

R: this information has been added to the revised text in the different sections.

Page 16 line 512, How many, male or female?

R: the sentence has been rephrased: Male and female TSSM (about 20 individuals per gender)

Page 17 line 551, How many? Leaf age?

R: The sentence has been rephrased in order to explain better the leaf age and type. The number of replicates has been indicated also for controls, as it was for treatments.

Page 17 line 557, Males or females or both and age?

R: ten adult females were used, the sentence has been rephrased

Page 17 line 562, Number of replicates?

R: A new sentence reminds that the number of replicates is always five.

Page 17 line 567, How many

R: the sentence has been rephrased in order to better clarify the quantity of TSSM used

Page 17 line 576, Just put the maximum number that was repeated Remember that the reader will want to repeat the experiment, so you must be as precise in the methodology.

R: the sentence has been rephrased in order to clarify the number of extracts pooled.

Page 18 line 603, Include cite

R: reference (DOI: 10.1007/s10529-014-1479-4) has been added as requested

Page 18 line 626, young or older leaves

R: cotyledonary and primary leaves has been added

Page 19 line 659, Phaseolus lunatus change to P. lunatus

R: done

Page 20 line 706 change and co-workers to et al.

R: done

Page 20 line 709, What data?

R: the sentence has been rephrased

Page 20 line 710, Clarify SD

R: we clarified standard deviation (SD)

Page 20 line 711, Tukey or Bonferroni?

R: this sentence has been rephrased

Page 20 line 714, clarify Resistant

R: the term resistant was replaced with resistance

Page 20 line 718, change to tetranychids

R: done

Page 20 line 721 remove s from TSSMs

R: done

Reviewer 2 Report

Thank you for the invitation to meet your manuscript titled: Biology of Two-Spotted Spider Mites (Tetranychus urticae): Ultrastructure, Photosynthesis, Guanine Transcriptomics, Ca- rotenoids and Chlorophylls Metabolism, and Decoyinine as a Potential Acaricide.

The study is very interesting and adds to the knowledge in the field and area of acarology. The manuscript is generally well-written, concise and succinct, yet quite detailed. The statistical analysis is well chosen, but needs to be supplemented and clarified information. The results are presented clearly. The material and methods were described in detail and used appropriately.

Please consider corrections:

- Why is material and methods point 4? and are in favor of discussion? Are these the requirements of the editors?

- Supplementary Table S1 and Supplementary Table S2 - The tables show the results of significance of differences (Tukey test – ANOVA). Why the use of the test is not stated under Statistical analysis (Material and methods). Information should be completed - in which case the Tukey test was used; I point out that the authors used the Bonferroni test equivalently.

-  Table 3. „….In the same row, different letters indicate significant (P < 0.05) differences; n.c., not computable.” There is an inaccuracy with the assumed probability level of 0.01 in Material and methods. The same applies to the interpretation of the results in Section 2.10. Information is missing for Tukey test (Material and Methods).

Please verify the smallest accepted alpha error: 0.01 or 0.05 – in Bonferroni test and Tukey test.

The same is true for Table 2 and Table 1, an inaccuracy from Material and Methods. Interpretation of significant differences in 2.8 and 2.7.

The same is true for Figure 5 and  Figure 7 an inaccuracy from Material and Methods. Interpretation of significant differences in 2.6 and 2.5.

- Figure 5 - what it shows on the graph – whiskers?

Why in material and methods under statistical analysis - no calculation of the standard error is given. This is not the same as the standard deviation.

- Figure 3 - if possible - resolution to be improved

- Supplementary Figure S1: resolution to be improved

- Supplementary Figure S2: resolution to be improved

Author Response

The study is very interesting and adds to the knowledge in the field and area of acarology. The manuscript is generally well-written, concise and succinct, yet quite detailed. The statistical analysis is well chosen, but needs to be supplemented and clarified information. The results are presented clearly. The material and methods were described in detail and used appropriately.

R: we thank very much the reviewer for the appreciation of our work

Please consider corrections:

- Why is material and methods point 4? and are in favor of discussion? Are these the requirements of the editors?

R: yes this is the style of the Journal

- Supplementary Table S1 and Supplementary Table S2 - The tables show the results of significance of differences (Tukey test – ANOVA). Why the use of the test is not stated under Statistical analysis (Material and methods). Information should be completed - in which case the Tukey test was used; I point out that the authors used the Bonferroni test equivalently.

R: we thank the reviewer for raising this issue. A new sentence in the materials and methods clarifies the use of Tukey test.

-  Table 3. „….In the same row, different letters indicate significant (P < 0.05) differences; n.c., not computable.” There is an inaccuracy with the assumed probability level of 0.01 in Material and methods. The same applies to the interpretation of the results in Section 2.10. Information is missing for Tukey test (Material and Methods).

R: we thank the reviewer for raising this issue. A new sentence in the materials and methods clarifies the use of Tukey test.

Please verify the smallest accepted alpha error: 0.01 or 0.05 – in Bonferroni test and Tukey test.

The same is true for Table 2 and Table 1, an inaccuracy from Material and Methods. Interpretation of significant differences in 2.8 and 2.7.

The same is true for Figure 5 and  Figure 7 an inaccuracy from Material and Methods. Interpretation of significant differences in 2.6 and 2.5.

R: we thank the reviewer for raising this issue. A new sentence in the materials and methods clarifies the use of Tukey test.

- Figure 5 - what it shows on the graph – whiskers?

Why in material and methods under statistical analysis - no calculation of the standard error is given. This is not the same as the standard deviation.

R: we thank the reviewer for noticing this typo. The term error has been replaced with deviation

- Figure 3 - if possible - resolution to be improved

R: we have expended the figure to improve resolution

- Supplementary Figure S1: resolution to be improved

- Supplementary Figure S2: resolution to be improved

R: being supplementary data they do not need a high resolution. In order to keep them in the same page we had to shrink the original figures. We hope the reviewer will understand.